# *Moesziomyces antarcticus MMF1* Has a Role in the Secretion of Mannosylerythritol Lipids

**DOI:** 10.3390/microorganisms13071463

**Published:** 2025-06-24

**Authors:** Yuze Xu, David T. Stuart

**Affiliations:** Department of Biochemistry, University of Alberta, Edmonton, AB T6G 2H7, Canada; yuze3@ualberta.ca

**Keywords:** *Moesziomyces antarcticus*, mannosylerythritol lipid, biosurfactant, *MMF1 transporter*

## Abstract

Mannosyl erythritol lipids (MELs) are glycolipid biosurfactants produced by *Ustilaginomycete* yeasts. The MEL biosynthetic pathway has been characterized in *Ustilago maydis* where a putative transporter encoded by *MMF1* is required for the secretion of the glycolipid surfactant to the extracellular space. The anamorphic yeast *Moesziomyces antarcticus* is a prolific producer of MELs, but the mechanism of MEL secretion is less well characterized than in *U. maydis*. Homologous recombination was employed to generate a disruption of the *MMF1* gene in *M. antarcticus* JCM10317. This mutation did not prevent the intracellular accumulation of MEL species but did result in significantly reduced secretion of the conventional MEL-A, MEL-B and MEL-C species detectable by thin-layer chromatography. However, the mutant strain did secrete a glycolipid species that is distinct from conventional MEL-A/B/C and similar to a glycolipid secreted by *MMF1* mutant strains of *U. maydis* and *Pseudozyma tsukubaensis*. Despite the defect in MEL secretion displayed by the *M. antarcticus* strain harbouring a disrupted *MMF1* gene, these cells did not display a significant defect in growth or cell morphology. The findings of this investigation provide evidence that *M. antarcticus MMF1* encodes a transporter required for the secretion of MELs but not required for MEL synthesis or cell growth.

## 1. Introduction

*Ustilaginomycetes* are a class of fungi that produce several compounds that are important for biotechnology applications, including ustilagic acid (UA) ergothioneine (EGT), and mannosylerythritol lipids (MELs) [1,2,3]. Mannosylerythritol lipids (MELs) consist of a hydrophilic sugar moiety, typically a mannosyl group linked to the sugar alcohol erythritol through a glycosidic linkage. The mannosyl group is acylated at the C2’ and C3’ with acyl chains of variable lengths. The mannosyl ring can be further modified by acetylation at the C4’ and C6’ positions. The diacetylated form is referred to as MEL-A, monoacetylated isoforms are referred to as MEL-B and MEL-C, and the unacetylated form is named MEL-D. (Figure 1) [4].

The enzymes required for MEL biosynthesis are encoded by a co-regulated gene cluster that was first discovered and characterized in *Ustilago maydis* in 2006 [5]. Key enzymes in this pathway were found to be encoded by *EMT1*, a glycosyl transferase responsible for the synthesis of the 4-O-ß-D-mannopyranosyl-erythritol carbohydrate component. *MAC1* and *MAC2* are peroxisome-localized acyltransferases that acylate the mannosylerythritol and *MAT1*, an acetyl transferase that catalyzes acetylation of the mannosyl-ring [5,6]. Gene deletion studies performed with the individual candidate genes revealed that cells harbouring deletions in any of *EMT1*, *MAC1*, or *MAC2* were unable to synthesize MELs [5,6]. Strains lacking a functional *MAT1* gene could produce only the unacetylated MEL-D isoform [5]. Within the MEL biosynthetic gene cluster, a fifth gene was identified whose amino acid sequence was predicted to have multiple transmembrane helices, with features representative of major facilitator superfamily transporters [5]. This putative transporter, named *MMF1*, was considered to be a candidate MEL transporter owing to its sequence similarity to transporters, the location of its gene within the MEL biosynthetic cluster, its induction by low nitrogen similar to *EMT1*, *MAC1, MAC2* and *MAT1* and its localization of the Mmf1-GFP fusion protein to the plasma membrane of *U. maydis* [5,6,7].

The MEL biosynthetic gene cluster is conserved in a variety of fungi that have been demonstrated to produce MELs [8]. While many of the genes in this cluster have been shown to be essential for MEL biosynthesis, the putative transporter *MMF1* has yet to be well characterized. The initial description of the *MMF1* gene in *U. madis* reported that the deletion of the gene resulted in a deficiency in MEL secretion consistent with a role for *MMF1* as a transporter. The MEL gene cluster of *Pseudozyma tsukubaensis* has a similar genomic arrangement to *U. maydis*, and alignment of the coding sequence of the MEL biosynthetic genes displays a high identity with *U. maydis* (71%). Wild-type *P. tsukubaensis* cells mainly secrete di-acylated MEL-B as their most abundant form of MEL [9]. A recent investigation of *MMF1* function in *P. tsukubaensis* demonstrated that a *∆mmf1* strain produced a reduced total MEL yield but surprisingly also produced extracellular MELs, with the majority product identified as low-hydrophobicity 2′ mono-acylated MEL-D, followed by 2′ mono-acylated MEL-B [9].

A more extensive analysis of *MMF1* function in *U. maydis* revealed that *∆mmf1* strains were defective in their secretion of acetylated MELs. The strain displayed a reduced production of monoacetylated MEL-B and MEL-C. In contrast, diacetylated MEL-A was produced but retained in the cells [10]. Inactivation of *MMF1* in both *P. tsukubaensis* and *U. maydis* results in the accumulation and secretion of a form of MEL not detected in strains with a functional *MMF1* [9,10]. This MEL is proposed to result from the degradation of the di-acetylated MEL-A that cannot be secreted from the *∆mmf1* strain [10].

Although *MMF1* is included in the MEL biosynthetic gene clusters for other MEL-producing strains, the function of this putative transporter has only been investigated in *U. maydis* and *P. tsukubaensis*.

The anamorphic yeast *Moesziomyces antarcticus*, previously classified as *Pseudozyma antarctica*, is a member of the *Ustilaginaceae* family [11]. *M. antarcticus* JCM10317 isolated from lake Vanda in Antarctica is a demonstrated MEL producer [12]. This halotolerant strain displays the ability to produce MELs from a wide variety of feedstocks including oils, glycerol, sugars or lignocellulosic materials [13,14]. The MEL biosynthesis gene cluster of *M. antarcticus* displays an organization that is distinct from either *P. tsukubaensis* or *U. maydis* [9]. The *P. tsukubaensis MMF1* displays a 72% amino acid sequence identity with *M. antarcticus MMF1* [9]. However, despite the similarity of the genes involved in MEL synthesis and secretion, these two organisms display differences in the MEL species produced and secreted [15]. When the *M. antarcticus MMF1* gene was expressed in the microalgae *Chlamydomonas reinhardtii*, the cells displayed increased cell size, the accumulation of storage lipid and chlorophyll content but there was no evidence of lipid or glycolipid secretion [16].

Owing to the limited understanding surrounding MEL biosynthesis and secretion by *M. antarcticus* coupled with the industrial value of MEL products, we have investigated the requirement for *M. antarcticus MMF1* in the production and secretion of MELs. For this investigation we report the effect of *MMF1* disruption on MEL production and secretion. We further report the effect of *MMF1* deletion on cell growth and morphology.

## 2. Materials and Methods

### 2.1. Microbial Strains and Culture Medium

*Moesziomyces antarcticus* JCM10317 acquired from American Type Culture Collection (ATCC 34888) was routinely propagated at 30 °C on Yeast Extract Peptone Dextrose (YEPD) agar plates (10 g/L of Yeast extract, 20 g/L of peptone, 20 g/L of glucose, and 15 g/L of agar). *Saccharomyces cerevisiae* BF264-15Du (*Mata ade1 bar1∆ his2 leu2 trp1-1 ura3∆*) [17] was routinely maintained on YEPD agar. *Escherichia coli* DH5α (*fhuA2 lac(del)U169 phoA glnV44 Φ80’ lacZ(del)M15 gyrA96 recA1 relA1 endA1 thi-1 hsdR17*) used for all cloning and plasmid propagation was maintained on lysogeny broth (10 g/L of Tryptone, 5 g/L of yeast extract, 10 g/L of NaCl, 15 g/L of agar). Plates and liquid medium were supplemented with antibiotics as required for plasmid maintenance. All media components and antibiotics were purchased from Fisher Scientific Canada, Ottawa, ON, Canada. All restriction enzymes used were purchased from New England Biolabs Canada, Whitby, ON, Canada.

### 2.2. Plasmid Construction

The *M. antarcticus* hygromycin resistance cassette was constructed by amplification of the Glyceraldehyde-3-Phosphate Dehydrogenase 1 (*GPD1*) promoter, *GPD1* terminator and Phosphoglycerate Kinase 1 (*PGK1*) promoter from *M. antarcticus* genomic DNA with oligonucleotide pairs VP15-P13H, T5-T3, and P25-P23V. The coding sequence of the *E. coli hph* gene encoding hygromycin resistance was amplified from plasmid pAG32 [18] using oligonucleotide pair P1H5-H3T. All oligonucleotide sequences used were purchased from Integrated DNA Technologies (IDT) Coralville, IA, USA and are listed in Table 1.

The purified Polymerase Chain Reaction (PCR) fragments were combined with vector YCplac33 [19] that had been digested with *Eco* RI and *Hind* III and used to co-transform an *S. cerevisiae* strain. The transformants were selected on synthetic complete medium lacking uracil (SC-ura) agar plates (0.67% Yeast nitrogen base without amino acids, without ammonium sulphate, 0.5% ammonium sulphate, 0.2% casamino acids, 2% dextrose, 1.8% agar) and then tested for hygromycin resistance on YEPD agar plates supplemented with 300 µg/mL of hygromycin B (ThermoFisher Scientific, Waltham, MA USA). Correct plasmids were confirmed by colony PCR [20] and isolated by transformation of *E. coli*.

The *M. antarcticus EMT1* gene was amplified from genomic DNA with primers EMTD5-EMTD3. The purified fragment was combined with plasmid YEplac195 [19] that had been digested with *Eco* RI and *Hind* III and assembled via Gibson isothermal assembly [21]. The cloned *EMT1* was digested with *Sma* I and the PCR-amplified hygromycin cassette was inserted to disrupt the *EMT1* open reading frame yielding plasmid *emt1::HYGRO*.

The *MMF1* disruption plasmid was constructed by amplification of 1100 bp of the 5′ *MMF1* open reading frame and 1000 bp of sequence downstream from the stop codon from *M. antarcticus* genomic DNA with oligonucleotide pairs VM1.15/M1.1G3 and GTM5/TM3V. The hygromycin cassette and *PGK1*-GFP were amplified from plasmid DNA with primer pairs PGPD5/PPGK3 and P2GFP5/GFP3. All four DNA fragments and vector YEplac195 were gel-purified, mixed in a 1:1 molar ratio and assembled by Gibson isothermal reaction [21].

A plasmid to integrate the wild-type *MMF1* gene was generated starting with plasmid pAG25 that encodes a nourseothricin resistance gene [18]. The vector was digested with *Bgl* II/*Nco* I and ligated with an 800 bp fragment from the *M. antarcticus* Phosphofructokinase 1 *(PFK1)* promoter amplified from genomic DNA with oligonucleotides MaPFKp5 GGCAGCAGATCTTTTCTGAAGACCGACCCCTC/MaPFKp3 GCAGCACCCATGGGGTGTTGGAGGTGGCTGG. This placed the NatMX coding sequence under the regulation of the Ma*PFK1* promoter. The plasmid was subsequently digested with *Sal* I and ligated with the *M. antarcticus MMF1* gene that had been amplified from genomic DNA as a 3053 bp *Xho* I fragment with oligonucleotides MMF1x5 TACACCCTCGATCAGCGC/MMf1x3 TTTCGGTGCTGCTGCG.

### 2.3. Yeast Transformation

Plasmid and fragment transformations of *S. cerevisiae* were performed by lithium acetate–PEG-mediated transformation as described in [22]. Transformation of *M. antarcticus* was performed similarly but no heat shock was used in the procedure.

### 2.4. Southern Blot Hybridization

Total DNA from *M. antarcticus* was isolated as described in [23]. A total of 5 µg of DNA was digested separately with *Sac* II, *Afl* II and *Xho* I, or left undigested and separated on 0.8% (*w*/*v*) agarose and transferred to a Magnagraph nylon membrane (Micron Separations Inc, Westborough, MA, USA) by capillary transfer. The DNA probes generated as PCR products from cloned *MMF1* DNA with oligonucleotides (Probe 1f/Probe 1r) and (Probe 2f/Probe 2r) were radioactively labelled with ^32^P-dCTP (PerkinElmer Scientific, Vaughan, ON, Canada by random priming [24]. Membranes were prehybridized (6× SSC, 5× Denhardts, 0.4% SDS, and 50 µg/mL of salmon sperm DNA) for 2 h before being incubated with a radioactive probe overnight. Membranes were washed at 68 °C in 2× SSC, 0.1% SDS and signals were detected by autoradiography. The membrane was stripped with probe stripping buffer (5 mM of Tris·HCl, pH of 8.0, 2 mM of EDTA, 0.1× Denhardt’s Reagent) for 2 h at 65 degrees, and was re-probed with another probe. All chemicals for hybridization and stripping buffers were purchased from Fisher Scientific Canada, Ottawa, ON, Canada.

### 2.5. Cultivation Conditions for MEL Biosynthesis

*M. antarcticus* was cultivated in 25 mL of YPED medium for 48 h at 30 °C with shaking at 200 rpm in 250 mL Erlenmeyer flasks. A total of 1 mL of the YEPD culture was then diluted into 25 mL of seed medium (1 g/L of Yeast extract, 3 g/L of NaNO_3_, 0.3 g/L of KH_2_PO_4_, 0.3 g/L of MgSO_4_, and 40 g/L of glucose). Following 24 h of incubation at 30 °C with agitation at 200 rpm, cells collected by centrifugation at 3000× *g* for 3 min and then resuspended in MEL production medium (1 g/L of Yeast extract, 3 g/L of NaNO_3_, 0.3 g/L of KH_2_PO_4_, 0.3 g/L of MgSO_4_). The carbon sources for fermentation, filter-sterilized 50% glucose, final concentration (4% *w*/*v*; 12% *w*/*v*), 50% glycerol, and final concentration (6% *v*/*v*) were added before fermentation at an initial optical density (OD_600_ = 0.5). The cultures were incubated for 7 days and cell densities were monitored every 24 h.

### 2.6. MEL Analysis

For monitoring MEL production, 750 µL of whole culture was extracted with an equal volume of ethyl acetate 2 times with vigorous shaking. The organic phase was collected and dried at room temperature. The crude MEL was resuspended in 100 µL of chloroform for thin-layer chromatography (TLC) analysis. For monitoring secretion, the cellular and medium fraction were separated by centrifugation and the MEL extraction was performed on the cell pellet and medium fraction separately. The enriched MEL extract was resuspended in chloroform and chromatographed on Silica gel 60 TLC (Millipore Sigma Canada, Oakville, ON, Canada) plates using a Chloroform:methanol:NH_4_OH solvent system in a 65:15:2 (*v*:*v*:*v*) ratio [25]. The MEL species on TLC plates were visualized by spraying with orcinol-H_2_SO_4_ reagent (0.1% orcinol, 5% H_2_SO_4_) and heating in the oven for 5 min at 110 °C [26]. All chemicals for TLC were purchased from Millipore Sigma Canada, Oakville, ON, Canada.

### 2.7. Microscopy

Cell morphology was analyzed by microscopy using a Zeiss Axioskop 2 microscope and 63 x Plan NeoFLUOR objective (Carl Zeiss Canada, Toronto, ON, Canada). Images were captured with a CoolSNAP HQ2 CCD camera (Huntington Beach, CA, USA) and images were processed and false-coloured using Image Pro PLUS 6.1. Cell samples were stained with Nile red as described in [27]. Microscopic analysis was performed on cell samples spotted onto agar pads prepared as described in [28]. Statistical analysis of differences in growth rates and lipid body accumulation were performed by unpaired *T*-test using an online *t*-test calculator, Prism version 10.5.0 graph pad https://www.graphpad.com/quickcalcs/ttest1/ (Dotmatics, Boston, MA, USA).

## 3. Results

Our initial attempts to transform *M. antarcticus* using a conventional lithium acetate–PEG-mediated procedure had relatively low levels of success. We attempted to employ hygromycin B at 300 µg/mL as a selective agent as has been used successfully with *P. antarctica* T34 [29]. However, this procedure yielded a high background of false-positive colonies. To improve the selection process, we tested hygromycin B sensitivity of *M. antarcticus* JCM10317 by plating 1 × 10^6^ early log phase cells onto YEPD plates supplemented with hygromycin B at concentrations ranging from 100 to 700 µg/mL. Inspection of the plates after 48 h of incubation revealed that a hygromycin concentration of 500 µg/mL completely stalled colony formation. Based on the observed sensitivity of the *M. antarcticus* strain, a concentration of 500 µg/mL of hygromycin was used for subsequent experiments. 

### 3.1. M. antarcticus MMF1 Inactivation by Homologous Recombination

To investigate the requirement for *MMF1* function in MEL synthesis and secretion, a gene disruption vector consisting of 1.1 kB of the 5′ end of the *MMF1* coding sequence from the start codon to position 1096, a hygromycin resistance cassette composed of the *E. coli* hygromycin phosphotransferase gene (*hph*) including the *M. antarcticus GPD1* promoter and terminator, a GFP open reading frame and the terminator region of *MMF1* (1047 bp of DNA downstream of the coding sequence) from wild-type *M. antarcticus* genomic DNA was constructed. Correct integration of this construct into the *M. antarcticus* genome was anticipated to delete 531 bp of DNA sequence corresponding to 177 amino acid residues from the *MMF1* coding sequence, replacing them with the hygromycin expression cassette (Figure 2A).

A culture of *M. antarcticus* was subjected to lithium acetate–PEG-mediated transformation and candidates were selected by growth on YEPD agar plates supplemented with 500 µg/mL of hygromycin B. The genomic structure of the candidates was screened by performing PCR with oligonucleotide pairs composed of one oligonucleotide primer that hybridized within the gene disruption cassette sequence and a second that hybridized either upstream (VM15/H3T) or downstream (P1H5/TM3V) of the deletion cassette sequence. Positive candidates yielding the expected PCR products from the initial screening were identified and subjected to a further analysis using Southern blot hybridization. Genomic DNA was isolated from a parental wild-type strain and from candidate *mmf1::HYGRO* disruptants and subjected to digestion with restriction enzymes *Sac* II, *Xho* I, *Afl* III or no enzyme. These genomic DNA were separated by agarose gel electrophoresis. Staining with ethidium bromide demonstrates that similar amounts of DNA were included in the uncut and digested samples (Figure 2B). The DNA was subsequently probed with a radio-labelled 300bp DNA fragment hybridizing with a coding sequence of *MMF1* that is expected to be deleted by the *mmf1::HYGRO* disruption vector (Figure 2C, Probe 1). While the wild-type *M. antarcticus* genomic DNA displayed the expected band pattern *Sac* II—3723bp, *Xho* I—3033bp, *Afl* III—3677 bp when probed with this fragment (Figure 2C, WT lanes), no signals can be detected from the samples of the candidate *mmf1::HYGRO* disrupted strain (Figure 2C, ∆ lanes). A second 300bp DNA probe was prepared from sequences 3′ to the *MMF1* coding sequences that are expected to be present in both the *MMF1* and *mmf1::HYGRO* strains (Figure 2A, Probe 2). Probing the genomic preparations with Probe 2 yielded bands for both *MMF1* (*Sac* II—3723bp, *Xho* I—3033bp, *Afl* III—3677bp) and *mmf1::HYGRO* (*Sac* II—2907bp, *Xho* I—6898bp, *Afl* III—1587bp) that are consistent with the genomic structure expected from the correct integration (Figure 2D). The combined PCR and Southern blotting results confirmed the isolation of an *M. antarcticus* strain with the *MMF1* gene disrupted by the HYGO cassette (*mmf1::HYGRO*).

### 3.2. Disruption of M. antarcticus MMF1 Alters MEL Secretion but Not Synthesis

After confirming the correct replacement of *MMF1* with the *mmf1:HYGRO* cassette, thin-layer chromatography was employed to investigate the impact of *MMF1* inactivation on the MEL production and secretion profile. The *MMF1* wild-type and isogenic *mmf1:HYGRO* strains were cultured in MEL synthesis medium supplemented with 4% glucose. Following three days of cultivation, equivalent samples were collected from each culture and the cell pellets were separated from the culture medium prior to the extraction of MELs. The MEL species separated by TLC were detected with orcinol staining. The wild-type *MMF1* strains (*MMF1*) displayed accumulation of MEL-A/B/C species in both the cell pellets and medium fractions consistent with the ability to secrete MELs into the extracellular space (Figure 3A). In comparison to the *MMF1* cultures, strains harbouring the *mmf1::HYGRO* allele (*mmf1∆*) displayed a pronounced reduction in MEL-A/B/C species in the medium fraction (Figure 3A, Med). Despite the lack of secreted MELs, the cell pellet samples collected from the *mmf1::HYRGO* strain displayed an orcinol staining spot on the TLC plate migrating at the same position as MEL-A, indicating that the *mmf1::HYGRO* cells remained capable of synthesizing MELs (Figure 3A, Cells). The total amount of MELs accumulated within the cell pellets was not visibly reduced by the *mmf1::HYGRO* genotype. In fact, the MEL-A spot in the cellular fraction of *mmf1::HYGRO* was slightly larger than that of the wild-type, indicating an accumulation of MEL-A within the cells (Figure 3A–C, Cells). We also observed a shift in MEL species in the cellular fraction of *mmf1::HYGRO*, where the spot representing MEL-C stained less intensely than that in the *MMF1* wild-type strain, suggesting that the absence of *MMF1* may impact the forms of MEL species that accumulate intracellularly (Figure 3A, Cells). Close inspection of the TLC plate revealed the presence of an orcinol staining spot that was enriched in the *mmf1::HYGRO* samples (Figure 3A, Med). This species, indicated with an asterisk, is not detectable or at least not prominent in the wild-type *MMF1* samples.

To confirm MEL synthesis by the *mmf1::HYGRO* strain, three biological replicates of strains *MMF1*, *mmf1::HYGRO* and *emt1::HYGRO* were cultured under MEL synthesis conditions using glucose as a carbon source. TLC analysis of cell pellet and culture medium fractions demonstrated the accumulation of MELs by the *MMF1* strains (WT) in both the cell pellets and medium fractions (Appendix A). As has been previously demonstrated, the deletion of *EMT1* (*emt1∆*) which catalyzes the first step in MEL synthesis resulted in a complete loss of detectable MELs in both the cell pellet and medium fractions (Appendix A). In contrast, cell pellets from the *mmf1::HYGRO* strains displayed enrichment for MEL-A but little of MEL A/B/C could be readily detected in the culture medium fractions (Appendix A). The orcinol staining spot that can be detected in the *mmf1::HYGRO* medium fraction is absent from the medium samples collected from the *emt1∆* strain, consistent with the notion that this does represent an MEL species enriched in the secreted fraction from strains lacking a functional *MMF1* gene (Appendix A).

MEL accumulation was then monitored in *MMF1* and *mmf1::HYGRO* strains cultured in medium supplemented with either 12% glucose (Figure 3B) or glycerol as a carbon source (Figure 3C). Following three days in culture, a MEL was produced by the cells in both carbon sources. However, in the *mmf1::HYGRO* samples, MELs could be extracted from the cell pellets but was nearly undetectable from the culture medium fractions of cells cultured in 12% glucose (Figure 3B, compare Med and Cell fractions). We did detect some MEL-A in the medium fraction of the *mmf1:HYGRO* strain cultured in glycerol-supplemented medium but it was much less abundant than the MEL-A in the medium fraction from the *MMF1* strain (Figure 3C). The absence or substantial reduction in the conventional MEL species in the medium fraction suggested that *MMF1* plays an important role for MEL secretion when cells are cultured in either glucose of glycerol-supplemented medium.

The requirement for *MMF1* in the secretion of the conventional MEL A/B/C species was confirmed by restoring *MMF1* function to the *mmf1::HYGRO* strain. Plasmid MMF1-NatMX carrying a full-length *MMF1* coding sequence, 989 bp of the *MMF1* promoter, 250 bp of 3′ terminator sequence and a nourseothricin resistance gene regulated by *M. antarcticus PFK1* promoter was used to transform *mmf1::HYGRO*. The genomic DNA from candidate nourseothricin-resistant transformants was tested by PCR to confirm that a full-length *MMF1* coding sequence had been introduced into the mutant strain. A strain harbouring the *mmf1::HYGRO* and *MMF1-NatMX* allele was cultured in MEL synthesis conditions. Both cell pellets and culture medium fractions were extracted and assayed for MEL production by TLC. The major MEL-A/B/C species were readily detectable in both the cell pellet and medium fractions in the *mmf1::HYGRO* strains harbouring an integrated full-length *MMF1* (Figure 3D, *∆ WT*). In comparison, the parent *mmf1::HYGRO* strain lacking the restored *MMF1* allele accumulated MEL A/B/C intracellularly but few MELs could be detected in the culture medium (Figure 3D). The observation that the loss of *MMF1* function results in deficient MEL secretion and the installation of a functional *MMF1* gene restores secretion provides support for the contention that *MMF1* is required for MEL secretion in *M. antarcticus.*

### 3.3. MMF1 Disruption Does Not Significantly Affect the Rate of Cell Growth in Culture

The growth profile of the *mmf1::HYGRO* strain was determined in MEL production conditions to assess whether the inability to secrete MELs and the intracellular accumulation of MELs affects cell proliferation. Wild-type cells, *emt1::HYGRO*, and *mmf1::HYGRO* strains were cultured in MEL synthesis media with three different carbon sources: 4% glucose (Figure 4A), 12% glucose (Figure 4B), and 6% glycerol (Figure 4C). The cell densities of triplicate cultures were monitored over six days. Overall, no significant difference in growth was observed between *mmf1::HYGRO* (Figure 4A–C, open circles) and *MMF1* cells (Figure 4A–C, filled circles) in any of the three carbon sources. In contrast, the *emt1::HYGRO* strain (Figure 4, open squares) grew slower than the other two strains in all tested conditions. When the overall growth rates were compared, it was found that *M. antarcticus* JCM10317 grew fastest in 6% glycerol, followed by 4% glucose and 12% glucose. The difference between the two glucose concentrations might be explained by osmotic pressure, which resulted in a longer lag-phase significant delay in the growth curve of the 12% glucose group before 96 h. These findings suggest that the absence of the *MMF1* gene does not significantly impact cell growth in culture conditions conducive to MEL production with different water-soluble carbon sources and that the *EMT1* gene may play a role in regulating growth in these conditions.

### 3.4. Lipid Droplet Accumulation Is Not Altered by Deletion of MMF1

The wild-type *M. antarcticus*, *MMF1* and *mmf1::HYGRO* strains were examined by microscopy to determine if the inactivation of *MMF1* induced any alteration in the cells’ morphology. Consistent with the observation that the mutation of *MMF1* does not reduce cell proliferation under MEL synthesis conditions, we did not observe any distinguishable differences in cell morphology between a wild-type *M. antarcticus* JCM10317 strain and isogenic *mmf1::HYGRO* (Figure 5A, DIC panels). We further investigated the morphology of the strains by staining neutral lipids with Nile red. Both strains displayed brightly staining masses consistent with the formation of lipid droplets (Figure 5A, Nile Red panels). Similar structures have previously been observed in MEL producers [30]. We observed some variation in the number of Nile red staining bodies per cell (2–4) (Figure 5B) but quantitative analysis detected no statistically significant difference between the two strains.

## 4. Discussion

The results of our investigation provide evidence consistent with the proposal that the protein encoded by *MMF1* is important for the effective secretion of the conventional MEL A/B/C species produced by *M. antarcticus*. Disruption of the *MMF1* open reading frame resulted in a reduction in MEL A/B/C that are detectable in the culture medium but allowed their intracellular accumulation (Figure 3A–C). Subsequent introduction of an *MMF1* coding sequence to the *mmf1::HYGRO* strain restored the secretion of MELs to the culture medium (Figure 3D). These findings support a role for Mmf1 in the secretion of MELs similar to the functions previously determined for *MMF1* encoded by *U. maydis* [10]. This observation contrasts with the report that the deletion of *MMF1* encoded by *P. tsukubaensis* does not prevent the secretion of the major MEL species into the extracellular space [9]. This difference may reflect a distinction in the molecular machinery responsible for the transport of MEL species across the plasma membrane and it may be that *P. tsukubaensis* encodes more than one protein capable of promoting this transport. The *U. maydis* Mmf1 has been localized to the plasma membrane whereas the *P. tsukubaensis* Mmf1 has not been localized to any specific cellular location and little is known of its functions or physical interactions with other proteins. The observed difference in MEL secretion displayed by the *mmf1* deletion strain of *P. tsukubaensis* in comparison with *M. antarcticus* shown in this work could also be attributed to the carbon sources used in the MEL production phase. We supplied only glucose or glycerol to *M. antarcticus* as a carbon source while the experiments reported for *P. tsukubaensis* employed olive oil. This may have created a form of selective pressure to induce MEL secretion by *P. tsukubaensis* or simply created a hydrophobic sink allowing some form of passive transport of the MELs into the medium. We observed some MEL-A in the medium of the *mmf1::HYGRO* strain cultured in glycerol-supplemented medium consistent with the idea that the culture medium may influence the secretion of some MEL species. A more extensive analysis of the enzymes’ catalytic activities will be required to clarify this.

MEL synthesis and intracellular accumulation persisted in the *M. antarcticus mmf1*::*HYGRO* strain, indicating that Mmf1 is dispensable for MEL biosynthesis. Interestingly, the defect in MEL secretion did not produce a significant effect on the cellular growth or gross morphology of *M. antarcticus*, with lipid droplet accumulation remaining unaltered (Figure 4 and Figure 5). It is expected that there would be a limit to how many MELs could accumulate within the cells before the surfactant induces disruption of membranes or internal structures. We did observe a reduction in total MEL accumulation in the *mmf1∆* strain relative to the *MMF1* strain (Figure 3A–C). Similarly, the deletion of *MMF1* resulted in a significant reduction in MEL accumulation in both *U. maydis* and *P. tsukubaensis* [9,10]. Some reduction in production may occur owing to a limitation in the ability to store the material intracellularly. One possible explanation is that feedback or product inhibition might be imposed to prevent the accumulation of a toxic load of MELs. While not unreasonable, there is currently no data to support the product inhibition hypothesis. A second possibility is that the accumulated acetylated intracellular MEL species in the *mmf1∆* strain are either degraded or undergo alternative metabolic processing. This hypothesis is consistent with our observation of a shift in MEL species towards more hydrophobic MEL-A in all three different cultivation conditions employed in this study (Figure 3A–C). One possible explanation for this shift is the accumulation of monoacetylated MEL-B/C in the cell, leading to increased substrate availability for the acetyltransferase Mat1 to interact and acetylate the MEL-B/C leading to increased abundance of the more hydrophobic MEL-A. This shift toward more hydrophobic MEL-A was not observed in the previous study on *P. tsukubaensis mmf1∆* [9]. This may be because the *P. tsukubaensis* strain only produces MEL-B and does not produce MEL-A, likely due to the specific activity of Mat1.

Deletion of *MMF1* in *M. antarcticus*, like *U. maydis* and *P. tsukubaensis*, resulted in the appearance of a species of MEL that is not prevalent in strains expressing a functional *MMF1* (Figure 3A–C). The new MEL species was detected both intracellularly and secreted into the growth medium. A combination of mass-spectrometry and NMR analysis identified the secreted MEL as mono-acylated MEL-B in *P. tsukubaensis* and mono-acylated MEL-A in *U. maydis* [9,10]. To date, no enzyme with MEL-specific esterase activity has been identified that could be responsible for the de-acylation of MELs but this is the simplest model for the appearance of a mono-acylated MEL in the *mmf1* deletion strains. We have not performed a structural analysis of the newly identified MEL species secreted by the *M. antarcticus mmf1* mutant, but its mobility during thin-layer chromatography is consistent with mono-acylated MEL-A. This suggests that a similar mechanism for MEL metabolism may be active in both *U. maydis* and *M. antarcticus*. The presence of a pathway for MEL metabolism in *M. antarcticus* is an important advancement both toward understanding the role of MELs in the biology of this microbe and for the industrial production of MELs as a bioproduct. The demonstration that MELs can serve as a carbon source for the producing organism reinforces the need to optimize feeding schedules for MEL production experiments to minimize product loss from assimilation and degradation. Additionally, this reveals a potential new route that can be exploited for the production of mono-acylated MEL species that may have desirable properties.

## 5. Conclusions

Disruption of the *MMF1* gene in *M. antarcticus* JCM10317 results in the defective secretion of conventional MEL-A, MEL-B and MEL-C species when the strain is cultured in a medium with glucose or glycerol as the carbon source. The conventional MELs accumulate intracellularly but little could be detected in the extracellular medium. However, an MEL species that is distinct from conventional MEL-A/B/C was detected in the culture medium of an *M. antarcticus* strain that lacked a functional *MMF1* gene. This secreted species was not detected in the culture medium of wild-type cells or cells that are fully deficient in MEL biosynthesis as a consequence of the deletion of the *EMT1* gene. Despite the defect in MEL secretion displayed by the *∆mmf1 M. antarcticus* strain, we observe no significant growth defect relative to wild-type cells when the strains are cultured in a medium with glucose or glycerol as a carbon source.

## Figures and Tables

**Figure 1 microorganisms-13-01463-f001:**
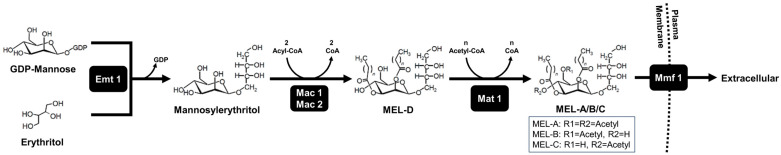
The proposed biosynthetic pathway for MELs. Formation of Mannosylerythritol with GDP-Mannose and erythritol, catalyzed by mannosyltransferase Emt1; transfer of acyl group to C2’ and C3’ of mannose ring, catalyzed by acyltransferases Mac1/Mac2; transfer of acetyl groups to C4’ and C6’ by acetyltransferase Mat1; export of MELs by putative exporter Mmf1.

**Figure 2 microorganisms-13-01463-f002:**
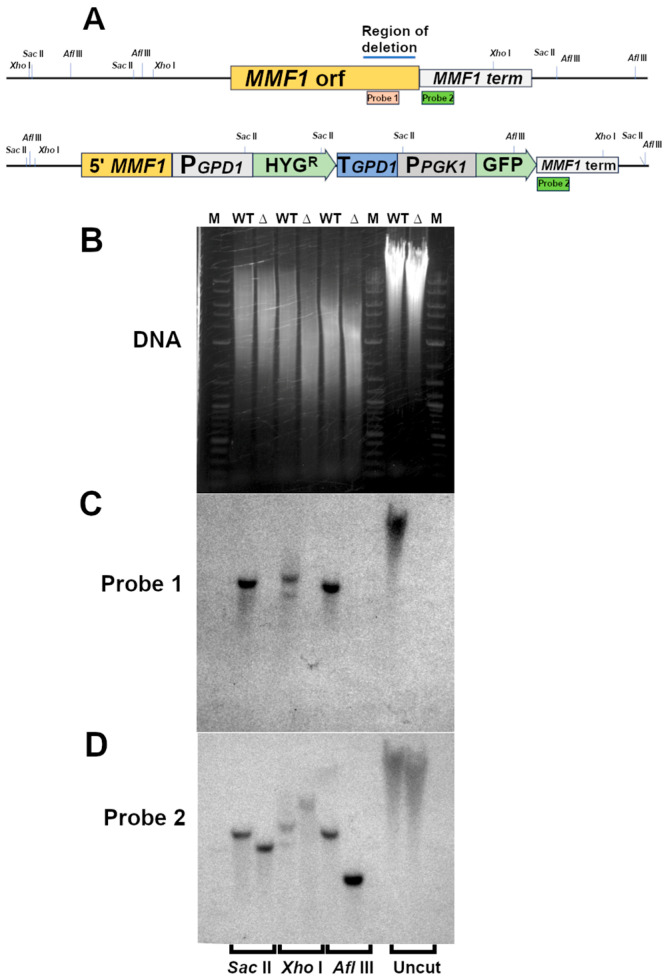
Genomic DNA analysis of *mmf1::HYGRO* mutation by Southern blot analysis. (**A**) Schematic diagrams of the wild-type *MMF1* gene, with the coding sequence displayed in orange and the 3′ untranslated region/terminator shown in grey. The sequence hybridizing with Probe 1 is indicated by the brown box (Probe 1) and sequences hybridizing to Probe 2 are indicated by the green box (Probe 2). The expected structure of the *mmf1::HYGRO* disruption also includes the relative positions of the inserted *GPD1* promoter (P*GPD1*), hygromycin B phosphotransferase coding sequence (HYG^R^), *GPD1* transcriptional terminator (T*GPD1*), *PGK1* promoter and Green Fluorescent Protein (P*PGK1* GFP). (**B**) Genomic DNA from *MMF1* and *mmf1::HYGRO* strains digested with the indicated restriction enzymes visualized by ethidium bromide staining (DNA). Molecular weight markers labelled M, DNA from *MMF1* strain is labelled WT, DNA from *mmf1::HYGRO* candidate is labelled ∆. (**C**,**D**) Autoradiography of a Southern blot of the same DNA probed with ^32^P labelled Probe 1 (**C**). The membrane was subsequently stripped and probed with Probe 2 (**D**).

**Figure 3 microorganisms-13-01463-f003:**
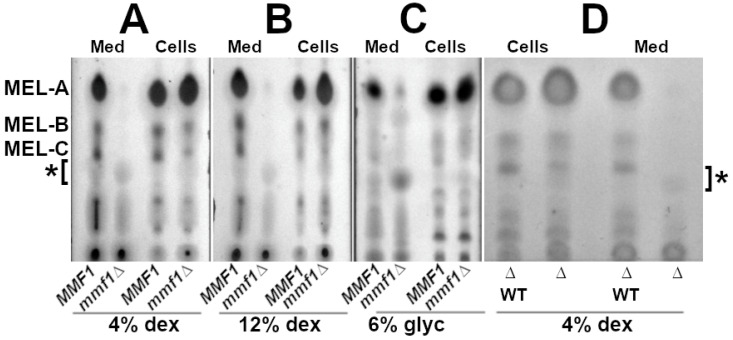
*MMF1* mutants are deficient in MEL secretion. MELs extracted from either culture medium (Med) or cell pellets (Cells) of cultures of *MMF1, mmf1::HYGRO* (*mmf1∆*) fractionated by TLC and visualized by orcinol staining. The major MEL-A/B/C species are indicated. An MEL species enriched in the *mmf1∆* strains is marked with an asterisk *. Strains were cultured in medium supplemented with (**A**) 4% glucose, (**B**) 12% glucose, and (**C**) 6% glycerol. (**D**) MELs extracted from the culture medium (Med) or cell pellet (Cells) of an *mmf1::HYGRO* strain transformed with a full-length *MMF1* coding sequence (*∆ WT*) or the parent *mmf1: HYGRO* strain (*∆*).

**Figure 4 microorganisms-13-01463-f004:**
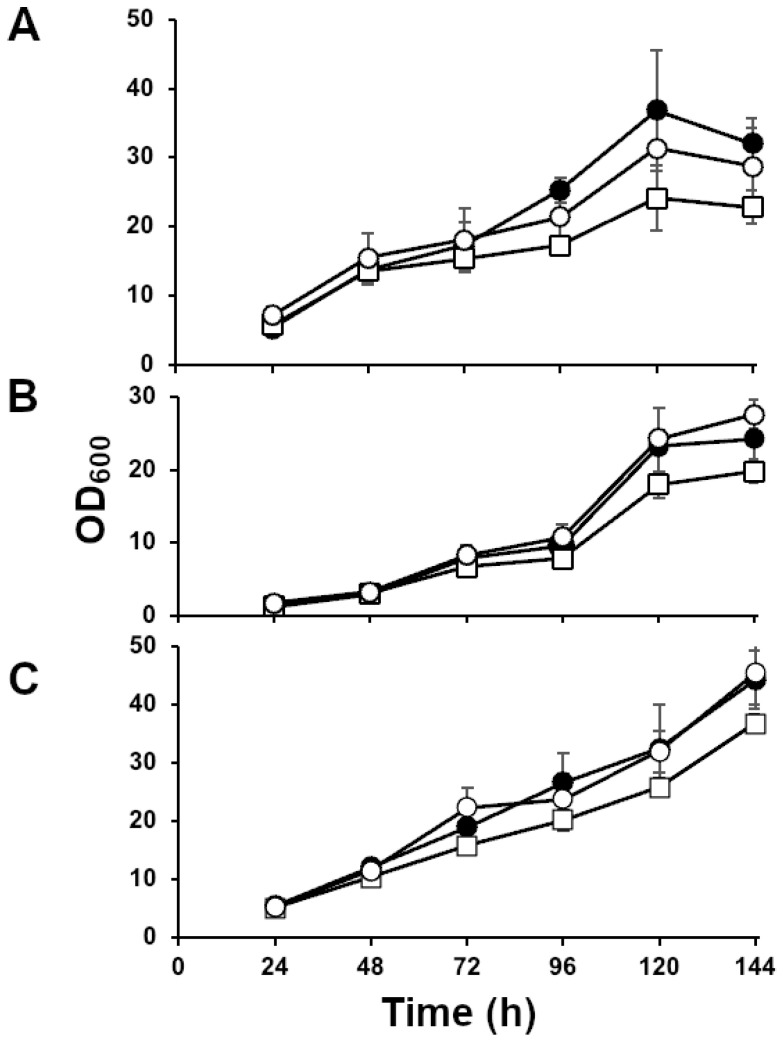
Deletion of *MMF1* does not reduce cell growth. Growth curves indicating culture density (OD_600_) over a six-day time course. *MMF1*, closed circles; *mmf1::HYGRO*, open circles; *emt1∆*, open squares. The strains were cultured in medium supplemented with the indicated carbon sources of (**A**) 4% glucose, (**B**) 12% glucose, or (**C**) 6% glycerol. Data shown are mean values of biological triplicate cultures; error bars indicate standard deviation.

**Figure 5 microorganisms-13-01463-f005:**
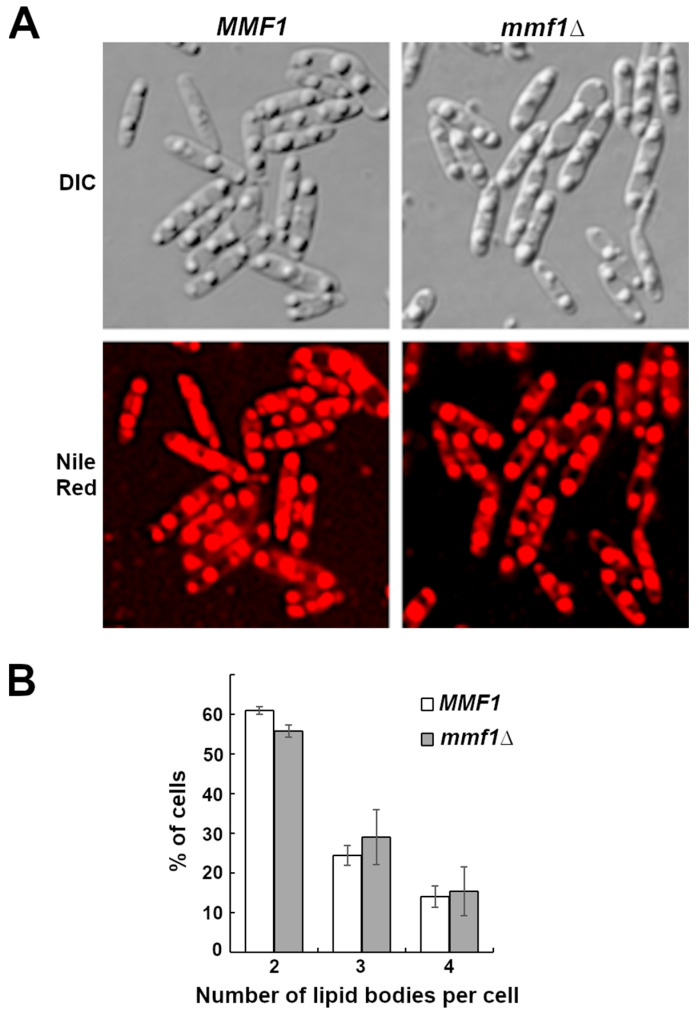
Deletion of *MMF1* does not alter morphology of actively growing cells. (**A**) Microscopic images of *MMF1* and *mmf1::HYGRO* strains observed under differential interference contrast (DIC) or epifluorescence to detect Nile red staining (Nile red). (**B**) The number of Nile red staining bodies (lipid bodies) per cell were counted and the number cells displaying 2, 3 or 4 staining bodies are indicated. Data are displayed as mean values from three independent cultures (*n* = 100 cells per culture). Error bars reflect standard deviation.

**Table 1 microorganisms-13-01463-t001:** Oligonucleotide primers used.

Primer Name	Sequence (5′—3′)
VP15	ACGACGTTGTAAAACGACGGCCAGTGAATTCGCCTCGGAAAGATCCTTCTGG
P13H	TTTTGAGATGATGGATGGGGAG
P1H5	GTGTCACACTCCCCATCCATCATCTCAAAAATGGGTAAAAAGCCTGAACTCAC
H3T	GGGGAGGGAGACGTAGGGAGCGTACTTATTCCTTTGCCCTCGGACG
T5	GTACGCTCCCTACGTCTCCC
T3	CAACTCCGCGGTATTTTGAGC
P25	GGCTCAAAATACCGCGGAGTTGCCACCTCGGCCGCG
P23V	TGACCATGATTACGCCAAGCTTAGTGGAGATGCGATCGTTTATCGG
M13f	GTAAAACGACGGCCAGT
M13r	CAGGAAACAGCTATGAC
EMTD5	GACGTTGTAAAACGACGGCCAGTGAATTCAGCGTACATAGGCTATTGAGC
EMTD3	GACTCTATGACCATGATTACGCCAAGCTTACTTCTGGTTCTACTGGTACC
EGR5	CTTCAAGCTCAACAAGCCGAAGCGCAACCCGCCTCGGAAAGATCCTTCTGGCTTTC
EGR3	TTGAGCGGCGAGGTGAATCCTGGACTGCCCCAACTCCGCGGTATTTTGAGCCC
ET5	AATGACAAAGTCCTCGCATCACGCATTCAC
ET3	GAATCTTTGACAAGCTCTTCGGAACAGTCATG
VM15	GTTGTAAAACGACGGCCAGTGAATTCTCGAGATGGACGACAAGATTGCGCTGACGA
M1G3	AAAGCCAGAAGGATCTTTCCGAGGCCTGATGATCAGTCCGGCGACCAGCG
PGPD5	GCCTCGGAAAGATCCTTCTGGCTTT
PPGK3	TGTGGAGATGCGATCGTTTATCGG
P2GFP5	CCCGATAAACGATCGCATCTCCACAATGAGTAAAGGAGAAGAACTTTTCAC
GFP3	CTATTTGTATAGTTCATCCATGCCATG
GTM5	TGGCATGGATGAACTATACAAATAGTCCGGTAGCTTTGGAGCCCT
TM3V	CTATGACCATGATTACGCCAAGCTTATGACCATTGTCAATGCCAAGACGC
IM5	CTGATCGCACTCGTGTTTG
IM3	CGCCAGTATCGAAGTTTGAG
Probe 1f	AACATCCAAAGCCGAGTGGG
Probe 1r	TTCGGCAGATACCTCAGCG
Probe 2f	AAATAGAAGGGAGGCGCTTG
Probe 2r	CAAGAGGGAGCTGGCCTTG
MaPFKp5	GGCAGCAGATCTTTTCTGAAGACCGACCCCTC
MaPFKp3	GCAGCACCCATGGGGTGTTGGAGGTGGCTGG
MMF1x5	TACACCCTCGATCAGCGC
MMf1x3	TTTCGGTGCTGCTGCG

## Data Availability

The original contributions presented in this study are included in the article/Appendix A. Further inquiries can be directed to the corresponding author.

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
