# Peer review of "Moesziomyces antarcticus MMF1 Has a Role in the Secretion of Mannosylerythritol Lipids"

_microorganisms, 2025, doi:10.3390/microorganisms13071463_

Round 1
Reviewer 1 Report
Comments and Suggestions for Authors
In this study, the authors employed homologous recombination to investigate the function of the MMF1 gene in M. antarcticus JCM10317. Their findings demonstrate that MMF1 deletion does not impair host strain growth but significantly reduces MEL secretion. These results suggest that MMF1 encodes a transporter specifically required for MEL secretion rather than for MEL biosynthesis or general cellular growth. The experimental evidence is compelling, and the conclusions are well-supported. The manuscript is clearly written, with logical flow and precise language. Prior to acceptance, the authors are suggested to address the following issues.
- The abbreviations in the text should be written out in full when they first appear. For example, in line 101, "YEPD agar plates" should be written as "yeast extract-peptone-dextrose (YEPD) agar plates." Similarly, on page 110, the full names of GPD1 and PGK1 should be provided: "GPD1 promoter (glycerol-3-phosphate dehydrogenase 1)" and "PGK1 promoter (phosphoglycerate kinase 1)."
- In line 224, the formatting of the restriction enzymes SacII, XhoI, and AflIII should be corrected to SacI I, Xho I, and Afl â…¢. Consistent revisions should be made for these enzymes throughout the manuscript.
- Line 331, the abbreviation "Fig. 4B" should be written in full as "Figure 4B" for consistency with the other figure references in this manuscript.
- Figure 5A: The current black-and-white image lacks sufficient resolution to clearly visualize Nile Red staining patterns. No distinct staining bodies are discernible in the presented image. I recommend replacing this with a high-resolution color image to enable proper evaluation of the staining results.
- Figure 5B: Both axes require proper labeling with clear indications of the represented parameters and their respective units to facilitate accurate interpretation of the data.
- Lines 393-394: “MEL synthesis and intracellular accumulation appeared to continue in the M. antarcticus mmf1::HYGRO strain indicating that Mmf1 is not required for MEL biosynthesis.” The original sentence is missing a comma before "indicating", and could be revised as: MEL synthesis and intracellular accumulation persisted in the M. antarcticus mmf1::HYGRO strain, indicating that Mmf1 is dispensable for MEL biosynthesis.
- Lines 395-396, “Interestingly the inability to secrete MEL did not cause a significant defect in cell growth or gross morphology of M. antarcticus including lipid droplet accumulation.”The phrasing of this sentence could be improved. Perhaps it would be better to revise it as follows: "Interestingly, the secretory deficiency of MEL did not significantly affect cellular growth or gross morphology in M. antarcticus, with lipid droplet accumulation remaining unaltered.
- Lines 405-407, “A second possibility is that accumulating acetylated intracellular MEL species in the mmf1∆ strain subject to degradation or undergo some other metabolic processing.” This sentence requires both grammatical and stylistic refinement. The following revised version might be more appropriate: “A second possibility is that the accumulated acetylated intracellular MEL species in the mmf1∆ strain are either degraded or undergo alternative metabolic processing.”
- In the Discussion section, the authors present results without citing the corresponding figure numbers, which may cause confusion. It is recommended to insert figure references at the end of relevant sentences, including the following instances: Lines 374-375, Lines 395-396, Lines 398-399, Lines 407-409, and Lines 416-418.
- Lines 412-413: "This shift toward more hydrophobic MEL-A was not observed in the previous study on P. tsukubaensis mmf1∆." It would be preferable to cite the reference supporting this comparative conclusion.

Author Response
Reviewer 1 Comments:
- The abbreviations in the text should be written out in full when they first appear. For example, in line 101, "YEPD agar plates" should be written as "yeast extract-peptone-dextrose (YEPD) agar plates." Similarly, on page 110, the full names of GPD1 and PGK1 should be provided: "GPD1 promoter (glycerol-3-phosphate dehydrogenase 1)" and "PGK1 promoter (phosphoglycerate kinase 1)."
Response: We thank the reviewer for careful reading of the manuscript. We agree with the reviewer's comment, therefore the full names for all abbreviations in the text have all been written out as suggested (lines 110 - 112, 117, 140 of the revised PDF).
- In line 224, the formatting of the restriction enzymes SacII, XhoI, and AflIII should be corrected to SacI I, Xho I, and Afl â…¢. Consistent revisions should be made for these enzymes throughout the manuscript.
Response: We agree with this comment, therefore we have changed all uses of restriction enzyme names in the revised PDF line 118, 127, 128, 140, 145, 146, 154, 229, 236, 241, 242 and throughout the manuscript have been changed to include italic and a space Sac I, Xho I, Afl III, including those incorporated into Figure 2, page 8 of the revised PDF as suggested by the reviewer.
- Line 331, the abbreviation "Fig. 4B" should be written in full as "Figure 4B" for consistency with the other figure references in this manuscript.
Response. We agree with this comment and therefore Fig, 4B was corrected to Figure 4B as suggested, now in line 340 of the revised PDF.
- Figure 5A: The current black-and-white image lacks sufficient resolution to clearly visualize Nile Red staining patterns. No distinct staining bodies are discernible in the presented image. I recommend replacing this with a high-resolution color image to enable proper evaluation of the staining results.
Response: We agree with this comment and therefore the original monochrome image has been modified to colour as suggested and a smaller frame of each microscope image was enlarged for ease of viewing as suggested by the reviewer (Figure 5A, page 12 of the revised PDF.
- Figure 5B: Both axes require proper labeling with clear indications of the represented parameters and their respective units to facilitate accurate interpretation of the data.
Response: We agree with this comment and therefore the axes of Figure 5B have been altered to include more informative names. The X-axis name has been changed to "Number of lipid bodies per cell" and the label "% of cells has been added to the Y-axis. These labels clarify the data and provide units to understand what is shown in the figure (Figure 5B, page 12 of the revised PDF).
- Lines 393-394: “MEL synthesis and intracellular accumulation appeared to continue in the M. antarcticus mmf1::HYGRO strain indicating that Mmf1 is not required for MEL biosynthesis.” The original sentence is missing a comma before "indicating", and could be revised as: MEL synthesis and intracellular accumulation persisted in the M. antarcticus mmf1::HYGRO strain, indicating that Mmf1 is dispensable for MEL biosynthesis.
Response: We agree with this comment and therefore the sentence, now line 406 - 407 of the revised PDF has been corrected to the language suggested by the reviewer.
- Lines 395-396, “Interestingly the inability to secrete MEL did not cause a significant defect in cell growth or gross morphology of M. antarcticus including lipid droplet accumulation.” The phrasing of this sentence could be improved. Perhaps it would be better to revise it as follows: "Interestingly, the secretory deficiency of MEL did not significantly affect cellular growth or gross morphology in M. antarcticus, with lipid droplet accumulation remaining unaltered.
Response: We agree with this comment and therefore the sentence, now line 407 - 408 of the revised PDF has been corrected to the language suggested by the reviewer.
- Lines 405-407, “A second possibility is that accumulating acetylated intracellular MEL species in the mmf1∆ strain subject to degradation or undergo some other metabolic processing.” This sentence requires both grammatical and stylistic refinement. The following revised version might be more appropriate: “A second possibility is that the accumulated acetylated intracellular MEL species in the mmf1∆ strain are either degraded or undergo alternative metabolic processing.”
Response: We agree with the comment and therefore the sentence, now line 418 - 420 of the revised PDF, has been corrected to the language suggested by the reviewer.
- In the Discussion section, the authors present results without citing the corresponding figure numbers, which may cause confusion. It is recommended to insert figure references at the end of relevant sentences, including the following instances: Lines 374-375, Lines 395-396, Lines 398-399, Lines 407-409, and Lines 416-418.
Response: We agree with the comment and therefore the sentences noted by the reviewer, now lines 383 - 384, 408 - 409, 412 - 413, 420 - 422, 429 - 430 of the revised PDF, have been corrected to include reference to the relevant figures as suggested by the reviewer.
- Lines 412-413: "This shift toward more hydrophobic MEL-A was not observed in the previous study on P. tsukubaensis mmf1∆." It would be preferable to cite the reference supporting this comparative conclusion.
Response: We agree with this comment and therefore the appropriate reference (Saika, et.el., 2020) has been added to the sentence, now line 426 - 427 of the revised PDF, as suggested by the reviewer.
Reviewer 2 comments:
I carefully read your manuscript, including interesting results of mannosylerythritol lipids production. I almost accepted your results and conclusion, but I had one unclear point in your results.
You mentioned that "The absence of the conventional MEL species in the medium fraction suggested that MMF1 is required for MEL secretion when cells are cultured in either glucose of glycerol supplemented medium" in line 306-308.
However, I could observe MELs in Med of HYGRO (mmf1Δ) containing 6% glycerol. Please explain the following point.
Response: We agree with the reviewer's comment. The reviewer makes an important point that we did not fully appreciate initially. We routinely are unable to detected secreted MEL-A when mmf1∆ cells are cultured in medium supplemented with glucose but some MEL-A can be detected in the medium fraction when the cells are cultured in medium supplemented with glycerol (as can be seen in Figure 3C, page 9 of the revised PDF). The amounts of MEL-A based on TLC analysis are much less than are observed with a wild-type MMF1 strain but still can be detected. Secreted MEL-A is also detected in Pseudozyma tsukubaensis cultures grown in medium supplemented with oil (Saika, et.al., 2020). A reasonable possibility, which we discuss is that oil or glycerol supplemented growth medium may either stimulate MEL secretion by an alternative route or may simply provide a "hydrophobic sink" allowing some MEL to escape the cells through a diffusion-like process. An important point made by the reviewer with which we agree, is that MMF1 seems not to be completely required for MEL secretion under all conditions. Therefore, in recognition of this we have softened several statements in the paper including the title to indicate that MMF1 has a role in MEL secretion rather than stating it is required (Manuscript title). The subtitle of section 3.2 (line 259 of the revised PDF) has been modified to reflect that MMF1 mutation alters MEL secretion rather than eliminating it. The appearance of MEL in the medium fraction of mmf1∆ strain cultured in glycerol medium is acknowledged (lines 311 - 314 of the revised PDF) and we indicate that mmf1 deletion reduces but does not fully eliminate MEL secretion in glycerol medium. The statement the reviewer refers to, now lines 311 - 314 has been revised to " The absence or substantial reduction of the conventional MEL species in the medium fraction suggested that MMF1 plays an important role for MEL secretion when cells are cultured in either glucose or glycerol supplemented medium. The possibility that growth medium conditions may alter MEL secretion is discussed in lines 401 - 405 of the revised PDF. We thank the reviewer for pointing this out.
Reviewer 2 Report
Comments and Suggestions for Authors
Dear Authors,
I carefully read your manuscript, including interesting results of mannosylerythritol lipids production. I almost accepted your results and conclusion, but I had one unclear point in your results.
You mentioned that "The absence of the conventional MEL species in the medium fraction suggested that MMF1 is required for MEL secretion when cells are cultured in either glucose of glycerol supplemented medium" in line 306-308.
However, I could observe MELs in Med of HYGRO (mmf1Δ) containing 6% glycerol. Please explain the following point.
Author Response
Reviewer 2 comments:
I carefully read your manuscript, including interesting results of mannosylerythritol lipids production. I almost accepted your results and conclusion, but I had one unclear point in your results.
You mentioned that "The absence of the conventional MEL species in the medium fraction suggested that MMF1 is required for MEL secretion when cells are cultured in either glucose of glycerol supplemented medium" in line 306-308.
However, I could observe MELs in Med of HYGRO (mmf1Δ) containing 6% glycerol. Please explain the following point.
Response: We agree with the reviewer's comment. The reviewer makes an important point that we did not fully appreciate initially. We routinely are unable to detected secreted MEL-A when mmf1∆ cells are cultured in medium supplemented with glucose but some MEL-A can be detected in the medium fraction when the cells are cultured in medium supplemented with glycerol (as can be seen in Figure 3C, page 9 of the revised PDF). The amounts of MEL-A based on TLC analysis are much less than are observed with a wild-type MMF1 strain but still can be detected. Secreted MEL-A is also detected in Pseudozyma tsukubaensis cultures grown in medium supplemented with oil (Saika, et.al., 2020). A reasonable possibility, which we discuss is that oil or glycerol supplemented growth medium may either stimulate MEL secretion by an alternative route or may simply provide a "hydrophobic sink" allowing some MEL to escape the cells through a diffusion-like process. An important point made by the reviewer with which we agree, is that MMF1 seems not to be completely required for MEL secretion under all conditions. Therefore, in recognition of this we have softened several statements in the paper including the title to indicate that MMF1 has a role in MEL secretion rather than stating it is required (Manuscript title). The subtitle of section 3.2 (line 259 of the revised PDF) has been modified to reflect that MMF1 mutation alters MEL secretion rather than eliminating it. The appearance of MEL in the medium fraction of mmf1∆ strain cultured in glycerol medium is acknowledged (lines 311 - 314 of the revised PDF) and we indicate that mmf1 deletion reduces but does not fully eliminate MEL secretion in glycerol medium. The statement the reviewer refers to, now lines 311 - 314 has been revised to " The absence or substantial reduction of the conventional MEL species in the medium fraction suggested that MMF1 plays an important role for MEL secretion when cells are cultured in either glucose or glycerol supplemented medium. The possibility that growth medium conditions may alter MEL secretion is discussed in lines 401 - 405 of the revised PDF. We thank the reviewer for pointing this out.
Round 2
Reviewer 1 Report
Comments and Suggestions for Authors
The authors have made necessary revisions to the original manuscript and responded to the reviewers' comments conscientiously. The paper's quality has been significantly enhanced, and I recommend accepting the revised version for publication.